# Comparative Studies of Polysialic Acids Derived from Five Different Vertebrate Brains

**DOI:** 10.3390/ijms21228593

**Published:** 2020-11-14

**Authors:** Yi Yang, Ryo Murai, Yuka Takahashi, Airi Mori, Masaya Hane, Ken Kitajima, Chihiro Sato

**Affiliations:** 1Bioscience and Biotechnology Center, Nagoya University, Chikusa, Nagoya 464-8601, Japan; yo.itsu0422@gmail.com (Y.Y.); murai.ryo@k.mbox.nagoya-u.ac.jp (R.M.); takahashi.yuka@i.mbox.nagoya-u.ac.jp (Y.T.); mori.airi@d.mbox.nagoya-u.ac.jp (A.M.); mhane@nuagr1.agr.nagoya-u.ac.jp (M.H.); kitajima@nuagr1.agr.nagoya-u.ac.jp (K.K.); 2Graduate School of Bioagricultural Science, Nagoya University, Chikusa, Nagoya 464-8601, Japan; 3Institute for Glyco-core Research (iGCORE), Nagoya University, Chikusa, Nagoya 464-8601, Japan

**Keywords:** polysialic acid, NCAM, brain evolution, native-PAGE, complex formation

## Abstract

Polysialic acid (polySia/PSA) is a linear homopolymer of sialic acid (Sia) that primarily modifies the neural cell adhesion molecule (NCAM) in mammalian brains. PolySia-NCAM not only displays an anti-adhesive function due to the hydration effect, but also possesses a molecule-retaining function via a direct binding to neurologically active molecules. The quality and quantity of polySia determine the function of polySia-NCAM and are considered to be profoundly related to the maintenance of normal brain functions. In this study, to compare the structures of polySia-NCAM in brains of five different vertebrates (mammals, birds, reptiles, amphibians, and fish), we adopted newly developed combinational methods for the analyses. The results revealed that the structural features of polySia considerably varied among different species. Interestingly, mice, as a mammal, possess eminently distinct types of polySia, in both quality and quantity, compared with those possessed by other animals. Thus, the mouse polySia is of larger quantities, of longer and more diverse chain lengths, and of a larger molecular size with higher negative charge, compared with polySia of other species. These properties might enable more advanced brain function. Additionally, it is suggested that the polySia/Sia ratio, which likely reflects the complexity of brain function, can be used as a new promising index to evaluate the intelligence of different vertebrate brains.

## 1. Introduction

Sialic acid (Sia) is an acidic nine-carbon sugar that shows structural diversity and consists of nearly 50 molecular species derived from *N*-acetylneuraminic acid (Neu5Ac), *N*-glycolylneuraminic acid (Neu5Gc), deaminoneuraminic acid (Kdn), and their modifications such as acetylation, sulfation, methylation, lactylation, and lactonization [1]. Polysialic acid (polySia/PSA) is a linear polymer of Sia that can be classified according to the degree of polymerization (DP), including diSia (DP = 2), oligoSia (DP = 3–8), and polySia (DP= 8–400). Inter-sialyl linkages such as α2,4-, α2,5-, α2,8-, α2,9-, α2,8/9- have been previously characterized [2]. In the context of the phylogenetic tree, the presence of polySia, according to biochemical analyses, was identified in organisms ranging from echinoderms such as sea urchins and sea cucumbers to vertebrates such as mice and humans. However, a considerable diversity in polySia is frequently observed, particularly in echinoderms. In contrast, in vertebrates, and particularly in the brain, only α2,8-linked di/oligo/polyNeu5Ac exists, and its evolutionary significance remains unknown. 

PolySia was first discovered in neuroinvasive bacterial glycocalyx as α2,8-linked polyNeu5Ac [3]. In vertebrates, it was first found in polysialoglycoprotein (PSGP) derived from salmonid eggs as α2,8-linked oligo/polyNeu5Gc [4], and later it was derived from neural cell adhesion molecule (NCAM) in rat embryonic brains [5]. In the brain, α2,8-linked polyNeu5Ac was shown to be present in the sodium channel α subunit in rats [6] and in SynCAM-1 (CADM1) in mice [7]. In humans and mice, milk CD36 has been demonstrated to be polysialylated [8]. Neuropilin-2 was also revealed to be a polysialylated glycoprotein in human dendritic cells [9]. In the brain, the majority of polySia was considered to attach to NCAM based on the observation that 90–96.5% of the polySia disappeared in NCAM knock-out (KO) mice [10,11], although several polySia-containing glycoproteins were shown as described above. Polysialylated NCAM (polySia-NCAM) is primarily expressed in embryonic and post-neonatal development brains and mostly disappears in adult brains, although the expression levels of NCAM remain unchanged [12]. In adults, polySia-NCAM expression is also observed in restricted areas where neural cells are actively generated, such as in the olfactory systems and the hippocampus [13,14]. Certain cells were also shown to be polysialylated in the amygdala, prefrontal cortex, hypothalamus, and other areas [15]. 

PolySia is a long, linear polymer that is considered to possess a helical structure. It forms a large “repulsive field” that is attributed to its hydration effect and the negative charge present at the carboxyl group at C1 of Sia [12,16]. NCAM undergoes cis- and/or trans-, homo- and/or heterophilic interactions with NCAM, adhesion molecules (L1, TAG-1), receptors (fibroblast growth factor receptor (FGFR), glial cell line-derived neurotrophic factor receptor alfa 1 (GFRα1)), and extracellular matrix components (collagen, heparan sulfate proteoglycan (HSPG), and chondroitin sulfate proteoglycan (CSPG)) [17,18]. When polySia links to NCAM, all the interactions between cell–cell and extracellular interactions that are described above are downregulated and the co-related intracellular signaling pathways are also downregulated [12]. Traditionally, this so-called “anti-adhesive” function of polySia was considered to be the only unique function of polySia. Recently, we have discovered the “reservoir function” of polySia that allows for the retention of soluble neurologically active molecules [19]. To date, brain-derived neurotrophic factor (BDNF) [20,21,22,23], fibroblast growth factor 2 (FGF2) [23,24], and catecholamine-based neurotransmitters such as dopamine, norepinephrine, and epinephrine [21,25] have been demonstrated to directly bind to polySia. Moreover, polySia-NCAM not only regulates the binding of these molecules to their receptors, but it also protects them from protease digestion [23]. Interestingly, polySia possessing different DP appears to exhibit a “preference” towards different molecules, as BDNF requires polySia possessing DP greater than 12 [20] and FGF2 [24] requires polySia possessing DP greater than 17. However, the underlying reason for the origin of polySia with different DP remains an unsolved question. The elongation of the polySia chain on NCAM is performed by two major polysialyltransferases (polySTs), namely ST8 alpha-*N*-acetyl-neuraminide alpha-2,8-sialyltransferase 2 (ST8SIA2, STX, siat8b) and ST8 alpha-*N*-acetyl-neuraminide alpha-2,8-sialyltransferase 4 (ST8SIA4, PST, siat8d). ST8SIA2 and ST8SIA4 can extend polySia chains alone; however, when coexisting in vitro, their effects have been suggested to be synergistic rather than additive [26]. Surprisingly, we found that polySia-NCAMs synthesized by ST8SIA2 and ST8SIA4 possessed different binding properties towards BDNF and FGF2 [27], thereby indicating that qualitative structural differences in polySia-NCAM are caused by ST8SIA2 and ST8SIA4. This is also important for the generation of functional differences in polySia-NCAM [19].

Abnormalities in polySia-NCAM derived from brains of individuals with mental disorders have been reported [28]. Schizophrenia is one such case. Schizophrenia is a chronic disease that originates within the nervous system, and its primary symptoms include delusions, hallucinations, thought disorders, and lack of motivation [29]. In schizophrenic brains, decreased levels of polySia-expressing neurons in the hippocampus were observed [30] and polySia-binding molecules such as BDNF and DA are physiologically active substances that are established as molecules which are profoundly involved in schizophrenia. Studies using gene-targeting mice revealed a relationship between polySia and mental disorders. For example, NCAM-deficient mice exhibit impaired contextual and sonic fear conditioning [31]. Abnormalities in polySia expression in the hippocampal dentate gyrus of *St8sia2*-deficient mice was also shown [32] and these mice exhibited schizophrenia-like phenotypes [33]. In humans, certain single nucleotide polymorphisms (SNPs) of *ST8SIA2* have been reported in patients with schizophrenia, and the promoter regions of (SNP-1 and SNP-3) were determined to be involved in schizophrenia in the Japanese populations [34]. SNP-7 (E141K) is a cSNP with a mutation in the ORF of a patient with schizophrenia reported by Arai et al. and has been well-characterized biochemically. The enzymatic activity in vitro and in cells and the structure and functions of polySia were impaired [21,27]. Based on this, polySia was shown to be highly regulated by a genetic factor. Additionally, environmental factors such as stress and medicine can alter polySia. Acute stress leads to reductions in both the quantity and quality of polySia in the olfactory bulb and prefrontal cortex [35], and treatment with anti-schizophrenia agents can consistently upregulate the expression of polySia in the PFC [36], a location in which altered polySia-NCAM expression has been observed in schizophrenia patients [37]. Taken together, these findings indicate that the quantity and quality of polySia are both highly controlled in the normal brain and in tissues. As a consequence of genetic background or environmental factors, deviation of polySia from the normal state may occur, ultimately resulting in functional changes in polySia that can lead to an increased risk for disease onset and poor prognosis. 

Hence, analysis of the quality and quality of polySia is important to understand the function of polySia, as this can contribute to determining new means of diagnosis and cure for brain diseases. However, biochemical analyses examining the mechanisms by which the quantity and qualitative structure of polySia are altered in different situations are not of high quality, particularly the studies that focus on the heterogeneity of the structure of this molecule. The reason is that, conventionally, the presence of polySia has been determined only by the presence or absence of reactivity with the anti-polySia antibody. In contrast, our laboratory has pioneered a novel method for determination of the qualitative differences in polySia in regard to length and molecular size [21,38,39,40]. In our latest research, we demonstrated an integrated analysis method that combined several conventional analytical methods and a newly developed method [41]. We have already demonstrated this method to be useful for distinguishing different types of polySia in mouse brains at different developmental stages.

In this study, to understand the polySia forms that are not fully characterized and that exist within several animal brains, we used the same integrated analytical method to examine the differences in quality and quantity of polySia derived from different vertebrate brains (mammals, birds, reptiles, amphibians, and fish). 

## 2. Results

### 2.1. Brain Size, Protein Amount, and Sialic Acid Amount

The brain weight, protein amount, and Sia amount of five vertebrate brains (mouse, chicken, turtle, xenopus and goldfish) were analyzed. We used ddY mice (*Mus musculus*) as a mouse model, white leg horn (*Gallus gallus*) as a bird model, soft-shelled turtle (*Pelodiscus sinensis*) as a turtle model, African crowed flog (*Xenopus laevis*) as a frog model, and small goldfish (*Carassius auratus*) as a fish model. The brain weight of the chicken was the heaviest among the analyzed brains, and the brain weights varied from 0.070 to 3.1 g (Figure 1A). The protein ratio to the brain weight was then calculated based on the protein concentrations (Figure 1B), and we found that the protein amount/brain weight was almost identical among the analyzed species (around 6%). We also analyzed the ratio of Sia amount/protein amount (Figure 1C), showing that fish possessed the highest level of Sia in brain among these species.

### 2.2. Western Blotting Analysis of polySia-NCAM

#### 2.2.1. SDS-PAGE/Western Blotting Analysis

PolySia is one of the unique glycan structures within the brain, and it has been demonstrated that polySia within the brain is related to cell adhesion, cell migration, neurogenesis, memory, and clock function. To compare the amounts of polySia among the brains of five vertebrates, we first analyzed the polySia structure by conventional SDS–PAGE/Western blotting using three anti-polySia antibodies (12E3, 735, and 12F8) (Figure 2A–C). The 12E3 antibody recognizes the anti-oligo/polySia structure with an intact non-reducing terminal end in the oligo/polySia (DP ≧ 5) [42], and the 735 antibody recognizes the helical motif of polySia structure with a pitch of more than 11 residues (DP ≧ 11) and the binding site of 735 scFv was shown to be triSia of the polySia structure in the internal polySia chains [43]. The epitope of 12F8 has not been defined; however, it was used in recent polySia-studies. Brain homogenates derived from mouse and xenopus exhibited strong staining in response to anti-polySia antibodies of the brains analyzed (Figure 2A–C). While a positive staining, the goldfish staining was the weakest among them, suggesting the presence of small amounts of polySia and/or short polySia chains in goldfish. The turtle brain exhibited stronger 735 staining than the chicken brain, while they showed similar intensity of 12E3 staining, thus suggesting that the DP of polySia chain in turtle brain might be longer than that in chicken.

#### 2.2.2. Native-PAGE/Western Blotting Analysis

Recently, we have demonstrated that native-PAGE analysis allows to differentiate various types of polySia-NCAMs that reflect the entire molecular weight (Mw), charge, and conformation. Therefore, we analyzed the homogenates derived from five different vertebrate brains by the native-PAGE (Figure 3A–C). We found strikingly different profiles of polySia-NCAM compared to those obtained by SDS-PAGE analysis (Figure 2A–C). The patterns of 12E3 and 735 staining were nearly identical; however, those of 12F8 staining were prominent in that the mouse brain only exhibited intense smears, although other vertebrate brains showed far less or no intensity. Notably, the strong and smear staining on the native-PAGE with any of the anti-polySia antibodies were characteristic of mouse brain homogenates. These results suggest that polySia structures in native conditions differ from those in denatured conditions on SDS-PAGE, and that more epitopes that can be recognized by the antibodies are exposed under denatured conditions. Other vertebrate brains did also exhibit smears at 240–480 kDa and at greater than 720 kDa which was well observed on the 12E3 blot. In fish brains, the 240–480 kDa staining was even greater than the 720 kDa staining, especially when detected using 12E3. Strong staining at 480–720 kDa was also observed using the 12F8 antibody in all the vertebrates except goldfish. 

#### 2.2.3. SDS-PAGE/Native-PAGE (S/N) MAP Analysis

To gain a more insight into the polySia profiles observed on the blots, we performed SDS-PAGE/Native-PAGE (S/N) Map analysis as described previously [41]. Briefly, the start and end points of the smear on the SDS-PAGE and those on the native-PAGE were plotted to obtain the linear line, and the slope value for the line was obtained as the qualitative index of polySia. Among the animals tested, we found that mice exhibited a completely different slope compared to those of other animals (Figure 4), indicating that polySia-NCAMs derived from mice were completely different than those from other animals. These data showed that polySia-NCAM in the brain was different not only in quantity (Figure 2 and Figure 3), but also in quality (Figure 4).

#### 2.2.4. ELISA Analysis

To determine the amounts of polySia in different animal brains, we used the enzyme-linked immuno sorbent assay (ELISA) method in the presence and absence of endo-*N*-acylneuraminidase (Endo-N) treatment. As shown in Figure 5, profiles of the relative intensity of the antibody binding against the animal brains tested were very similar among the three antibodies. Notably, these profiles were more similar to those obtained by the native-PAGE (Figure 3), compared with those obtained by SDS-PAGE (Figure 2). The ELISA results might reflect the amount of polySia more in the native state than in the denatured state.

### 2.3. Chemical Analysis of polySia-NCAM

#### 2.3.1. Mild Acid Hydrolysis–Fluorometric Anion-Exchange Chromatography Method (MH–FAEC, Oligo-Analysis)

DMB is a specific fluorescent reagent that targets α-keto acids and is applicable for the fluorescent labeling of sialic acids. DMB labeling of the reducing terminus of oligo/polySia chains obtained by mild acid hydrolysis of polySia and subsequent separation of the labeled oligo/polySia on anion-exchange chromatography depending on the degree of polymerization (DP) (MH-FAEC) was first developed by us [39]. Based on this method, we observed the maximum DP of the polySia structure released under mild acid conditions, although the longest DP in the sample may not be precisely defined due to the instability of the polySia chain to mild acidic conditions used for the labeling [2,39]. MH-FAEC of colominic acid yielded comb-like peaks with various DPs, and the maximum DP detected was 39. Under the same conditions, the supernatant obtained by the ethanol precipitation of crude homogenates of vertebrate brains was analyzed, and the comb-like profiles were obtained (Figure 6). The maximum DPs for mouse, chicken, turtle, xenopus, and goldfish were 25, 14, 17, 16, and 11, respectively. The mouse brain possessed the longest polySia chains and goldfish brain did the shortest. These data were nearly consistent with those obtained by Western blotting (Figure 2 and Figure 3).

#### 2.3.2. Fluorometric C_7_/C_9_ analysis

We previously developed a fluorometric C_7_/C_9_ analysis to detect α2,8-linked oligo/polySia structures [38]. This method was applied to the PVDF membrane that α2,8-linked oligo/polySia glycoproteins were blotted after α2-3,6-sialiase treatment. We clearly detected both C_7_-Neu5Ac and C_9_-Neu5Ac derivatives from all the brain samples (Figure 7A). The detection of C_9_ showed the presence of α2,8-linked oligo/polySia chains, and the average DP were calculated based on the equation “(C_9_+C_7_)/C_7_” (Figure 7B). It should be noted that these values are not the real average DP, because they should be different if the Sia residues are attached on multiantennary *N*-glycans, capped by some modifications, or linked by other than α2,8-linkage. The relative average DP of the polySia of mouse brain was the highest and that of goldfish was the shortest (Figure 7B). The results were again consistent with those of MH-FAEC (Figure 6) and Western blotting (Figure 2 and Figure 3). 

### 2.4. Chromatographical Analysis of polySia-NCAM 

#### 2.4.1. Anion-Exchange Chromatography of polySia–NCAM and NCAM

To understand the features of the entire polySia-NCAM, we analyzed the net negative charge (NNC) of polySia-NCAM using anion-exchange chromatography. The samples were applied onto the DEAE-Sephadex A-25 column and eluted in a stepwise manner at 0.2–3 M NaCl in the presence of detergent. The fractions were analyzed by immunostaining using 12E3 (Appendix A left), and the elution profiles are shown in Figure 8A. In mouse, the majority of polySia-NCAM was eluted at 0.6 M NaCl (67%), and minor but significant amounts of polySia-NCAM were eluted at 0.8 M NaCl (18%) and 1.0 M NaCl (12%). Interestingly, even at 3.0 M NaCl, polySia-NCAM was also observed (3%). In chicken, the majority of polySia-NCAM (61%) was eluted at 0.4 M NaCl, although 39% were eluted at 0.6 M. For turtle brains, almost all polySia-NCAM was eluted at 0.6 M, suggesting that the turtle polySia-NCAM is homogeneous. PolySia-NCAM of xenopus brains showed the similar elution profile to that of mouse, although the amount of polySia-NCAM at higher NaCl concentrations in xenopus was smaller than that in mouse. In goldfish brain, the elution pattern resembled that of chicken brain, although the amount of the 0.6 M NaCl fraction was larger in xenopus than in goldfish. As a whole, mouse brain is concluded to contain polySia-NCAM of the largest NNC among the animal brains tested. 

To gain further insight into the NNC of polySia-NCAM, the endo-N-treated samples were subjected to the same anion-exchange chromatography. Endo-N treatment degrades polySia moieties to DP = 2–4. These fractions were analyzed using an anti-NCAM antibody (Appendix A right). Interestingly, endo-N-treated NCAM was eluted at 0.4 M and 0.6 M in all animals tested, except for turtle. In turtle brain, all the endo-N-treated NCAM was exclusively eluted at 0.4 M. Considering that endo-N-treated NCAM was eluted at 0.4 M and/or 0.6 M NaCl for all the samples tested, endo-N-digested polySia chains are suggested to contribute to the high-salt elution of polySia-NCAM at 0.6–3 M NaCl (estimated DP is 30 or greater). Additionally, mouse was indicated to possess longer polySia chains because the polySia-NCAM was eluted at 0.8 M and greater (33% of the total polySia-NCAM). Conversely, chicken and goldfish are shown to possess shorter polySia chains on NCAM (DP~4) (61% and 31%, respectively), because the polySia-NCAM eluted at 0.4 M (DP~4) occupied relatively large part of total polySia-NCAM. The average negative charge of polySia-NCAM derived from vertebrate brains is shown in Appendix A. To examine the presence of keratan sulfate (KS) on NCAM, we analyzed the 0.4 M and 0.6 M fractions of endo-N-treated samples using the anti-KS antibody 5D4. In mouse and chicken brains, we observed the 196-kDa gp, and in xenopus and goldfish brains, we observed aggregated components between the separating and stacking gels (Appendix A). We did not observe any bands in the turtle samples during analysis of the 0.4 M NaCl fractions. Thus, it is shown that NCAM was separated into 0.4 M and 0.6 M NaCl fractions depending on the absence or presence of KS, and that KS-containing glycans might be eluted mostly at 0.6 M-2 fraction. In chicken, the NNC was smaller due to the shorter polySia chains, which are the main component of total polySia-NCAM.

#### 2.4.2. Gel Filtration

To understand the structural features of native polySia-NCAM of five different vertebrate brains, we performed Sephacryl S-500 gel filtration chromatography. The elution profiles were monitored by native-PAGE/immunostaining using 12E3 (Appendix A). The polySia staining was categorized into three based on the molecular size, i.e., HMW for more than 720 kDa, MMW for 480–720 kDa, and LMW for lower than 480 kDa. Since the polySia-NCAM is usually migrated at around 250 kDa as a monomer in SDS-PAGE (Figure 2), these three molecular species HMW, MMW, and LMW can be inferred as complex forms of polySia-NCAM, thus tentatively designated as oligomer-polymer, dimer-trimer, and monomer, respectively. The elution profiles of the three molecular species derived from different vertebrate brains are shown in Figure 9. The elution peak of HMW, MMW and LMW of most of the vertebrates can be observed at fraction No. 20, 22, and. 24, respectively. According to the calibration curve, the molecular mass of fractions No. 20., 22, and 24 were supposed to be 2000, 510, and 130 kDa. Elution of HMW bands commenced at No. 12, and the peak was No. 18 in mouse brain. In contrast, the peaks of chicken, turtle, and xenopus brains were observed at No. 20. The peak for goldfish was No. 22. The size of the polySia-NCAM complex derived from mouse was the highest, and that from goldfish was the lowest. The molecular size of polySia-NCAM in HMW of mouse brain is much higher, compared to other vertebrates. MMW were the major polySia-NCAM complex, and the peaks of MMW were eluted between No. 20 and 22. Those of LMW were eluted at No. 22–26, which were corresponded to the monomer of polySia-NCAM. A summary of the molecular mass of the polySia-NCAM is presented in Appendix A. The difference between the molecular size estimated by native-PAGE and that estimated by gel filtration may arise due to the dissociation of the complex via electrostatic interactions.

## 3. Discussion

Our study suggests that mammals and avians possess larger brains than those of reptiles, amphibians, and fish. Generally, brain weight increases depending on the body weight, which is known as the brain-body allometric relationship. In an evolutionary context, the deviations of this allometric relationship may indicate the higher recognition abilities and a large brain in comparison with body size is referred to as cerebralization or encephalization, and the EQ (Encephalization quote) value, EQ = K × [brain weight]/[body weight]^3/4^, is used as the index of brain size [44,45,46]. This formula was firstly introduced by Jerison in 1973 [47], and it was fixed by Pilbeam and Gould in 1974 [48], and Martin in 1981 from a different aspect [49]. We adopted the idea of the latter one, based on the actural and large cross-species data sets [46]. First, we analyzed the brain weight and (protein amount)/(brain weight) of five vertebrate brains, and found that the brain weights differed among the vertebrates tested. The values of [protein amount]/[brain weight] derived from all tested animals were almost identical, thereby indicating that the protein ratio in brain weight was the same among the brains tested. In contrast, the Sia content per protein was different among the brains tested. The Sia content was high in goldfish and xenopus which are living in water, and it was almost identical content in three other animals. The most striking difference among vertebrate brains tested was observed in polySia staining in the native-PAGE analysis.

We established the index [relative intensity of polySia-stained band]/[Sia content], which can be used to indicate the concentration ratio of Sia to the polySia structure. Strikingly, the patterns of the index (Figure 10B) were nearly identical to those of the EQ value (Figure 10C). Sia is an important glyco-module that is involved in fertilization, development, inflammation, and neural activity. The ratio of Sia to polySia epitope clearly indicates the accumulation of Sia onto polySia, which is indicative of the effective usage of Sia for these biological/neurological activities. Therefore, this index may provide an index for intelligence. As the intelligence is a vague word and there are several definitions for it. By Roth and Dick, it was defined as the speed and success of how animals solve problems to survive in their natural and social environments, including problems in feeding, spatial orientation and social behavior [46]. The intelligence is achieved by the organized brain functions in the animals, including recognitions, learning and memory, and organization of the information to survive for life. All these higher brain functions are related to the normal and orchestrated organization of the brain areas, such as prefrontal cortex [50], amygdala [51] and hippocampus [52], where polySia expression is usually maintained [15,19], but impaired in the patients of mental disorders. The polySia expression may reflect the higher brain functions. Recently, the polySia and polySia biosynthesizing enzyme ST8SIA2 have been demonstrated to exhibit a relationship with mental disorders, such as schizophrenia, bipolar disorder, depression, and autism. Mental disorders refer to a disease of the human brain activity, and impairment of polySia may lead to higher brain function. Therefore, the polySia/Sia index may provide an index for intelligence. Additionally, the slope of the S /N map revealed a striking difference in mouse polySia-NCAM among the vertebrates tested. Both indices are useful for easy analysis of the quality of polySia-NCAM. Another merit of using the index about polySia is that even using three individuals, the differences of mammals compared to the other vertebrates could be apparently shown, which is also beneficial from the animal welfare perspective. Furthermore, the brain intelligence and recognition ability are also correlated with the cortex and prefrontal cortex [46], and it may be interesting to compare the two indexes, using the different brain regions related with different functions, such as cognition, learning, memory, and emotion.

For the close analyses, we applied mild acid hydrolysis-fluorometric anion-exchange chromatography (MH-FAEC) and fluorometric C_7_/C_9_ analyses, whose results show the minimum value of the maximum DPs (maxDP) and the average DP, respectively. Based on the results of maxDP, polySia chains from mouse are the longest and those from goldfish are the shortest, of those from all vertebrates tested. On the other hand, interestingly, the average DP of the majority of the polySia structures is almost identical among five different vertebrates. The difference between the maxDP and average DP may reflect the diversity in the DP size of polySia chains, and based on the data of the difference, polySia chains of mouse, turtle, and xenopus brains exhibit greater diversity in DP than those of chicken and goldfish brains. The biological meaning of the DP of polySia is demonstrated to reside in the molecule-binding properties of polySia. For example, polySia can bind to BDNF with a DP greater than 12. In contrast, a DP of greater than 17 for polySia is required for polySia to bind to FGF2 effectively. The longer polySia chain is highly effective for the binding of neurological and biologically important factors such as BDNF, FGF2, and dopamine, and the diversity of DP may allow polySia chains to participate in diverse brain activities. Therefore, it can be considered that mouse polySia, compared with polySia of other vertebrate brains, is highly functional. 

For the net negative charge (NNC) of polySia-NCAM that our study first focused on, it is interesting to note that the polySia-NCAM from mouse, chicken, xenopus, and goldfish brains is eluted at 0.4 M and 0.6 M NaCl on the anion-exchange chromatography, while the turtle brain polySia-NCAM is exclusively eluted at 0.4 M NaCl, indicating that, of those vertebrates tested, turtle expresses polySia-NCAM of the lowest NNC. As NCAM is modified with the KS, the turtle is suggested to possess a less KS-modified NCAM that is eluted at 0.4 M. With regard to reptiles, there exist two types of species, the turtle and the lizard, and it is worthy to compare the amount of KS-modified polySia-NCAM from lizard and turtle brains. On the SDS-PAGE/Western blotting using the anti-NCAM antibody 0B11, mouse NCAM shows discrete bands, while turtle NCAM shows smears (Appendix A). 0B11 is thought to recognize the extracellular regions of NCAM near the membrane, and it is thus possible that structural differences of this region of the mouse and turtle NCAM may affect the KS-modification of NCAM. The striking feature of mouse polySia-NCAM is its elution at 0.8 M, 1.0 M, and 3.0 M on the anion-exchange chromatography, thus indicating the presence of extremely high NCC. This property of mouse polySia-NCAM is due to the long polySia chain, because the polySia can be completely removed by endo-N treatment. The same phenomenon is also observed in adult pig brain homogenates [40]. The extremely high NNC is concluded to be characteristic in mammals. On the other hand, in goldfish and chicken brains, polySia-NCAM are only eluted at 0.4 M and 0.6 M NaCl fractions, indicating the presence of shorter polySia chains than in mammals. This conclusion is also suggested by the MH-FACE results. In addition, it is interesting to point out that the KS-modified polySia-NCAM is more abundant in chicken and goldfish than in mouse, because the goldfish and chicken polySia-NCAM remains in 0.4–0.6 M NaCl fractions even after Endo-N treatment. It should be noted that adult pig brains, 8 w and retired mouse brains tend to show more KS-modified polySia-NCAM than embryonic brains [40,41]. The function of KS-modified polySia-NCAM needs to be further studied. 

To our surprise, on the gel filtration chromatography, mouse contains more HMW-polySia-NCAM, compared with other vertebrate species. The molecular mass of HMW in mouse brain is estimated to be more than 2000 kDa, which suggests that polySia-NCAM is present as a complex of tetramer~octamer or even higher oligomers. Several lines of evidence show the complex formation of polySia-NCAM. First, the polySia-NCAM binds to polySia-NCAMs themselves, as demonstrated by the surface plasmon resonance-based method [27]. Interestingly, the polySia-NCAM synthesized by either of ST8SIA2 and ST8SIA4 can associate with polySia-NCAM synthesized by ST8SIA4, but not by ST8SIA2, and the complex formation is suggested to depend on the action of the polySTs. Second, polySia-NCAM is shown to form a large complex with FGF2 and BDNF (2000–5000 kDa) [2]. Third, polySia-NCAM also form a complex with proteoglycans, such as heparan sulfate proteoglycan (HSPG) and chondroitin sulfate proteoglycan (CSPG) through the Ig2 domain [2,18]. It should be noted that goldfish polySia-NCAM has the lowest molecular mass, being eluted at No. 22–26, which contains only the monomer or dimer. In addition, it is also interesting that xenopus polySia-NCAM has almost an equal amount of LMW and MMW, while mouse polySia-NCAM tends to have more MMW than LMW. Thus, the complex formation of polySia-NCAM is suggested to be dependent on the vertebrate species. Our finding that polySia-NCAM exists as a complex with itself or other molecules may have functional meanings, and further studies are necessary to solve the problem.

## 4. Materials and Methods 

### 4.1. Materials

Bovine serum albumin (BSA), anti-NCAM antibody (0B11) for detecting mouse, turtle and xenopus NCAM, α2-3,6-sialidase from *Clostridium perfringens* (Recombinant, *E. coli*, Cat. No. 480708), and trifluoroacetic acid (TFA) were purchased from Sigma-Aldrich (St. Louis, MO, USA). Anti-polySia antibody, 12E3, which recognizes the oligo/polyNeu5Ac structure (DP ≥ 5) [42], was generously provided by Dr. Tatsunori Seki (Tokyo Medical University, Tokyo, Japan). 735-ScFv, which recognizes the polyNeu5Ac structure (DP ≥ 11) [53], was purified as described previously [43]. Endo-N, which cleaves the oligo/polySia structure (DP ≥ 5) [54], was generously provided by Dr. Frederic A. Troy (University of California, Davis, CA, USA). 12F8 (purified rat anti mouse CD56, antigen: Mouse membrane fraction), which is considered to recognize the neuraminidase-sensitive epitope of N-CAM used as anti-polySia antibody, was purchased from BD Pharmingen (Franklin Lakes, NJ, USA). Anti-NCAM antibody (H300) for detecting chicken NCAM was purchased from Santa Cruz Biotechnology (Santa Cruz, CA, USA). Anti-NCAM antibody (123C3) for detecting fish NCAM was purchased from Abcam (Cambridge, UK). POD-labeled anti-mouse IgG + IgM and anti-rabbit IgG were purchased from American Qualex (San Clemente, CA, USA). Colominic acid, α2,8-linked polyNeu5Ac (average DP = 43), which is chemically and immunologically identical to the polySia structure in NCAM and phenylmethylsulfonyl fluoride, was purchased from Wako (Osaka, Japan). The PVDF membrane was obtained from Millipore (MA, USA). Enhanced Chemiluminescence Western blotting Detection Reagent, Sephacryl S-500 and DEAE-Sephadex A-25 were obtained from GE Healthcare (Buckinghamshire, UK). 1,2-dimethylenedioxybenzen (DMB) was purchased from Dojindo Molecular Technologies, Inc. (Kumamoto, Japan). Pre-stained Mw marker was obtained from Bio-Rad (Hercules, CA, USA). Anti-HSC70 antibody, NativeMark Unstained Protein standard, and BCA protein assay reagents were obtained from Thermo Fisher Scientific (Waltham, MA, USA). 

### 4.2. Animals and Ethics Statement

Mice (*Mus musculus*, ddY, male, adult) were maintained in a controlled environment (23 ± 2 °C and 50 ± 10% humidity, 12:12 light/dark cycle) with food and water available ad libitum. All procedures were approved by the Animal Care and Use Committee of Nagoya University (Permit Number: 2016022506). Frogs (*Xenopus laevis*) (adult) were obtained from Chubu Kagaku Shizai (Nagoya, Japan). Brain of chicken (*Gallus gallus domesticus*) (adult) was from Dr. Yasunari Ohmori (Nagoya University, Nagoya). The golden fish (Anekin, *Carassius auratus*) (adult) was from Pet plus (Nagoya, Japan). The head of Suppon (*Pelodiscus sinensis*) (adult) was from Yamato Yoshoku (Taku, Japan). We have adopted the 3R principles (Replacement, Reduction, and Refinement) and test plans were formulated with maximum consideration for minimizing pain on the requisite minimum number of animals, with respect for the lives of the animals, taking animal welfare into consideration. We used minimum animal number (*n* = 3) necessary for statistical analysis.

### 4.3. Sample Preparation

All brains were collected by surgery. The brains were homogenized with lysis buffer (1% Triton X-100, 1 mM phenylmethanesulfonyl fluoride, protease inhibitors: 1 μg/mL aprotinin, 1 µg/mL leupeptin, 1 µg/mL pepstatin, 2 µg/mL antipain, and 10 μg/mL benzamidine), 1 mM EDTA, 50 mM NaF, 10 mM β-glycerophosphate, 10 mM sodium pyrophosphate, and 1 mM sodium *o*-vanadate in PBS. The homogenates were incubated on ice for 1 h and centrifuged at 9600× *g* for 20 min at 4 °C. The supernatant was collected. Protein concentrations were measured using the BCA assay.

### 4.4. Content of Sialic Acid

Sia was measured as previously described [38,44]. Briefly, samples (25 µg protein as BSA) were diluted with an equal volume of PBS and hydrolyzed with 0.1 NTFA at 80 °C for 2 h. The samples were then dried by speed vac. To the dried samples, equal volumes of 0.01 NTFA and DMB solution were added and incubated at 50 °C for 2 h. The authentic Neu5Ac sample was used for evaluation. DMB derivatives were separated by the Wako Handy ODS column (250 × 4.6 mm, Wako, Osaka Japan).

### 4.5. Western Blotting

First, 10 µg of protein from each sample was separated by SDS-PAGE (7% acrylamide gel) or native PAGE (7% acrylamide gel) and proteins were blotted on PVDF membrane. The membrane was then blocked with 1% BSA with PBS containing 0.05% Tween 20 (PBST) at 25 °C for 1 h for anti-glycan antibody (12E3 and 735) or 2% skim milk for anti-protein antibody and 12F8 antibody. The membrane was incubated with the primary antibody. After washing with PBST, the membrane was incubated with secondary antibody, peroxidase-conjugated anti-mouse IgG + M antibody (1/5000 dilution) at 37 °C for 1 h. Primary antibodies used were anti-polySia antibody; 12E3 (1.0 µg/mL, mouse IgM), 735 (0.8 µg/mL, mouse IgG), 12F8 (0.5 µg/mL, rat IgM), loading control; anti-HSC70 antibody (0.28 µg/ ml, rabbit IgG), and anti-NCAM antibody (0B11, 4 µg/mL, mouse IgG) for detecting NCAM of mouse, turtle and xenopus, H300 (0.5 µg/mL, rabbit IgG) for detecting NCAM of chicken, and 123C3 (1 µg/mL, mouse IgG) for detecting NCAM of fish. 0B11 was reported to [55]. 

### 4.6. SDS-PAGE/Native-PAGE Map

To make the SDS-PAGE/Native-PAGE MAP, first, polySia-staining obtained in SDS-PAGE or Native-PAGE/Western was changed to pseudo color, and standard band was set as the origin of the map (in this case, mouse brain homogenate). Then, the distance from the origin to the top of the other band on SDS-PAGE (X_1_) and that on Native-PAGE (Y_1_) was calculated. Additionally, the distance from the origin to the bottom of the other band on SDS-PAGE (X_2_) and that on Native-PAGE (Y_2_) was calculated. Two points, A (X_1_, Y_1_) and B (X_2_, Y_2_), were plotted on a coordinate axis and linked with each other to make a correlation curve, y = K x + B. All curves from different samples were plotted on the same map and the slope (K) can be compared to reflect the qualitative differences of the polySia from each sample.

### 4.7. ELISA Analysis

Samples were adjusted to 5 µg/mL (as BSA) with PBS (the concentration of Triton X-100 should be below 0.03%) and 50 µL of the solutions was absorbed onto a 96-well immunoplate, and other procedures were performed as described [41].

### 4.8. Mild Acid Hydrolysis-Fluorometric Anion-Exchange Chromatography Analysis

Samples (100 µg protein as BSA) in 200 µL of 0.005N TFA were added to 200 µL of DMB solution and incubated at 50 °C for 1 h [39,40]. Then NaOH was added to a final concentration of 0.1 N, and the solution was then incubated at 37 °C to remove lactonization. After centrifugation at 15,000 rpm for 3 min at 4 °C, the supernatant was collected. One N HCl was added for neutralization, and the samples were diluted and subjected to an anion-exchange chromatography column (DNApac PA-100, 4 × 250 mm, DIONEX, Sunnyvale, CA, USA) and separated on a JASCO HPLC system. The sample was loaded on a column and eluted with 2 mM Tris-HCl (pH 8.0), followed by NaCl gradient (0–15 min, 0 M; 15–185 min, 0 → 0.6 M; 185–190 min, 0.6 → 1.0 M) in 2 mM Tris-HCl (pH 8.0) and 0.1% triton X-100. The flow rate was 1 mL/min and fractions were monitored with a fluorescent detector (Em 373 nm, Ex 448 nm, FP-2020, JASCO, Tokyo, Japan).

### 4.9. Chemical Analysis of α2,8-Linked Oligo/polySia Chains on Glycoproteins Blotted onto PVDF Membranes

Brain glycoproteins (20 µg protein as BSA) in 14 µL were added to 4 µL of 5× reaction buffer and 2 µL of α2-(3,6)-sialidase treatment (25 µU) and incubated at 37 °C for 1 h to release monoSia residues. The sialidase-treated samples were blotted onto PVDF membranes as described previously and areas above 100 kDa were cut out. The membranes were analyzed by the fluorometric C_7_/C_9_ method for internal sialyl residues, as previously described [2,38].

### 4.10. Analysis of polySia-NCAM from Brain Homogenates using Anion-Exchange Chromatography

Untreated or Endo-N-treated brain samples (500 µg BSA) were applied to DEAE-Sephadex A-25 anion exchange chromatography (1 mL) and FT with PBS containing 0.1% triton X-100 was obtained. Then, samples were eluted with the 0.2, 0.4, 0.6, 0.8, 1.0, and 3.0 M NaCl in 2 mM Tris-HCl (pH 8.0) with 0.1% Triton X-100. After the samples were subjected to SDS-PAGE and blotted onto a PVDF membrane, the amounts of polySia and NCAM in each fraction were determined by anti-polySia staining (12E3) or anti-NCAM antibody staining of the membrane. Endo-N treated sample for NCAM staining and intact sample for polySia-staining were used. The column was calibrated with colominic acid (authentic polySia). 

### 4.11. Analysis of polySia-NCAM from Brain Homogenates Using Gel-Filtration Chromatography

Intact brain samples (2 mg of the sample) were applied onto the Sephacryl S-500 (14 mL) and eluted with PBS containing 0.1% triton X-100. After the samples were electrophoresed with native-PAGE and blotted onto a PVDF membrane, the amounts of polySia in each fraction were determined by anti-polySia staining (12E3). The column was calibrated with NativeMark unstained Protein Standard (Thermo Fisher Scientific, Waltham, MA, USA).

### 4.12. Data Analysis

All values were expressed as the mean ± SE (n is three) and *p*-values were evaluated by the Student’s *t*-test in Figure 1, Figure 2, Figure 3, Figure 4 and Figure 5, Figure 7 and Figure 10. The value of each vertebrate was compared with that of mouse brain. 

## 5. Conclusions

We analyzed polySia-NCAM by several chemical and immunochemical methods and found that polySia-NCAM showed different properties from vertebrate to vertebrate (Figure 11). The quality and quantity of different types of polySia-NCAM observed among different animals might be important for specific functions in each animal.

## Figures and Tables

**Figure 1 ijms-21-08593-f001:**
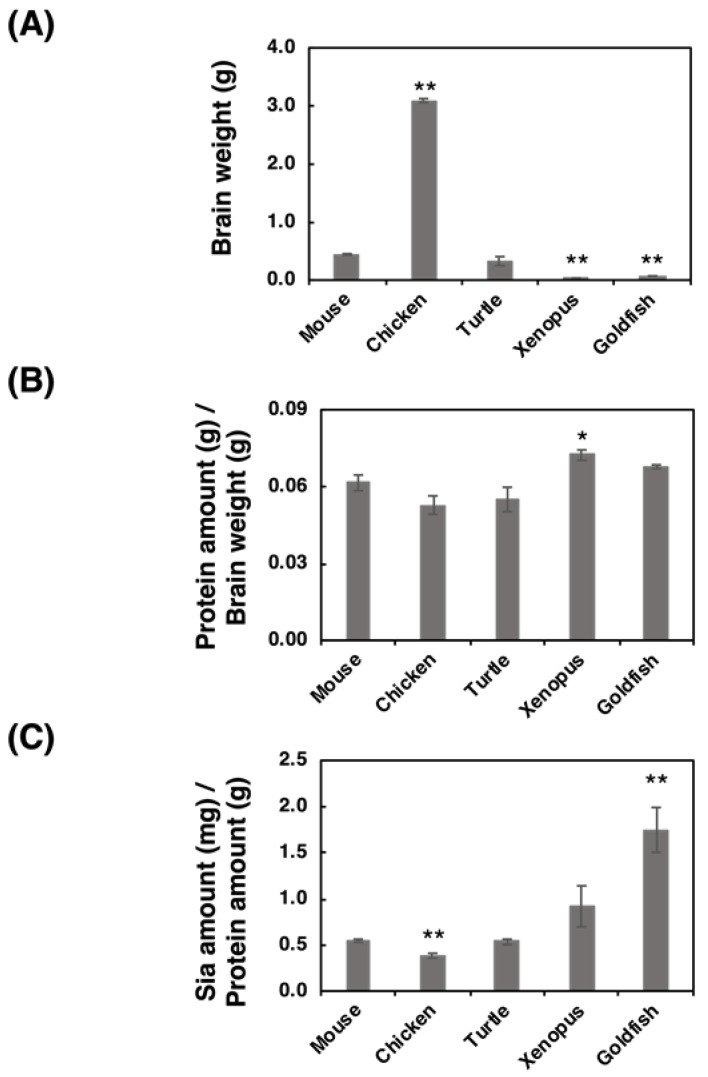
Comparison of brain weight, protein concentration, and Sia concentration for the five different vertebrate brains. Brains from five different vertebrates (mouse, chicken, turtle, xenopus, goldfish, (*n* = 3)) were used. (**A**) Comparison of the brain weights of the five vertebrate brains. (**B**) Protein amount (g) per brain weight (g). Brains were homogenized using lysis buffer, and a bicinchoninic acid (BCA) assay was performed to determine the protein concentration. (**C**) Sia amount (mg) per protein amount (g). Brain homogenate was treated with a strong acid to achieve hydrolysis of sialylglycoconjugates. All Sia residues were completely released and labeled with 1,2-dimethylenedioxybenzen (DMB). Sia–DMB was separated using a Wako Handy octadecylsilyl (ODS) column (250 mm × 4.6 mm, Wako). The absolute amount of Sia was calculated based on the authentic Neu5Ac. * indicates *p* < 0.05. ** indicates *p* < 0.01.

**Figure 2 ijms-21-08593-f002:**
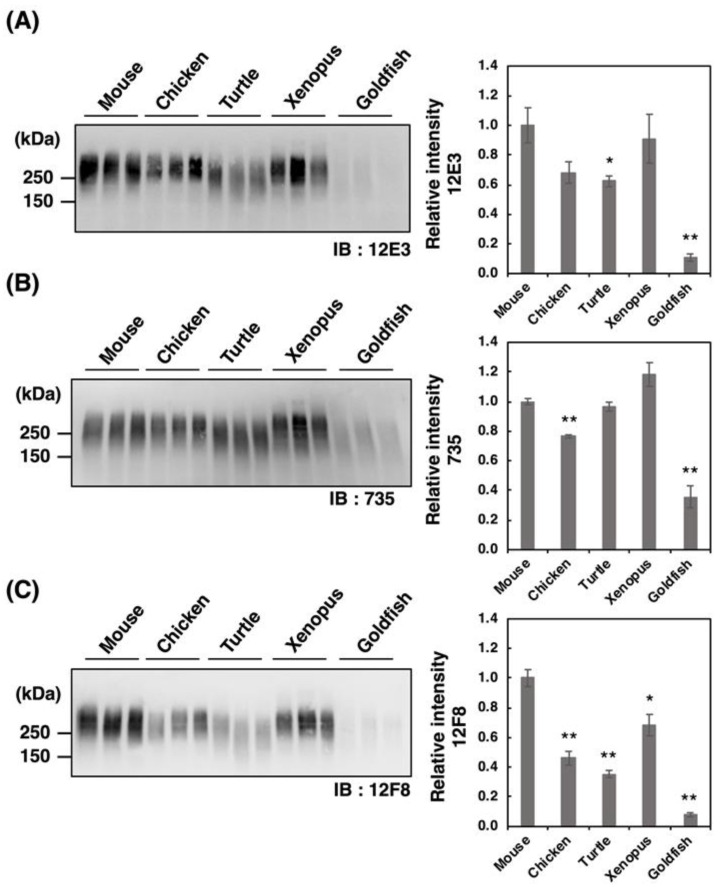
SDS–PAGE/Western blotting of polySia–NCAM derived from five vertebrate brains. Brains from five vertebrates (mouse, chicken, turtle, xenopus, goldfish, *n* = 3) were used. PolySia expression was analyzed using SDS-PAGE/Western blotting. The brain homogenates were separated by SDS-PAGE (7% polyacrylamide gel) and blotted onto polyvinylidene difluoride (PVDF) membrane (10 µg as protein/lane). The polySia-NCAM was visualized using anti-polySia antibodies, (**A**) 12E3, (**B**) 735, and (**C**) 12F8. The left panel represents the immunoblots. The right panel shows the relative intensity of the anti-polySia antibody. Mouse staining was set to 1.0. * indicates *p* < 0.05. ** indicates *p* < 0.01.

**Figure 3 ijms-21-08593-f003:**
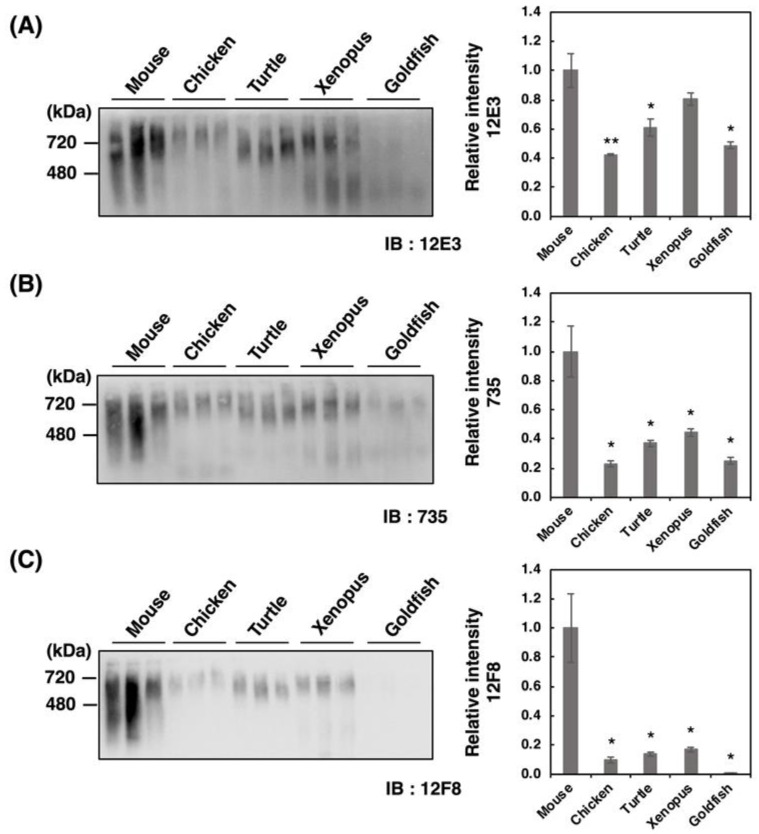
Native-PAGE/Western blotting of polySia-NCAM in brains from five different vertebrates. Brains from five vertebrates (mouse, chicken, turtle, xenopus, goldfish, *n* = 3) were used. PolySia expression was analyzed using native-PAGE/Western blotting. The brain homogenates of five vertebrate brains were separated by native-PAGE (7% polyacrylamide gel) and blotted onto PVDF membranes (10 µg as protein/lane). The polySia-NCAM was visualized using anti-polySia antibodies, (**A**) 12E3, (**B**) 735, and (**C**) 12F8. The left panel represents the immunoblots. The right panel shows the relative intensity of the polySia antibody. Mouse staining was set to 1.0. * indicates *p* < 0.05. ** indicates *p* < 0.01.

**Figure 4 ijms-21-08593-f004:**
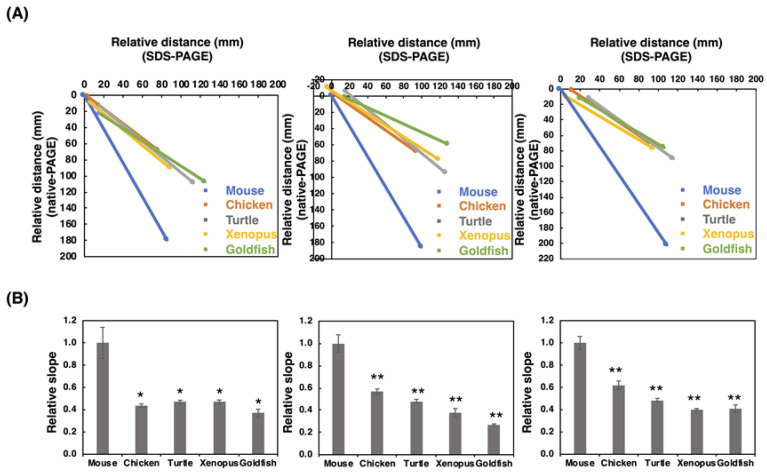
SDS-PAGE/native-PAGE MAP. The S/N MAP of brains from five different vertebrates is described based on the results of blots with each antibody. The results of SDS-PAGE (Figure 2) and native-PAGE (Figure 3) were used. (**A**) The typical S/N map of 12E3, 735, and 12F8 staining. (**B**) Relative slope value, K. The value for mouse was set to 1.0. The left, middle, and right panels stand for 12E3, 735, and 12F8, respectively. The values were obtained from the S/N maps from five different vertebrate brains (n = 3). * indicates *p* < 0.05. ** indicates *p* < 0.01.

**Figure 5 ijms-21-08593-f005:**
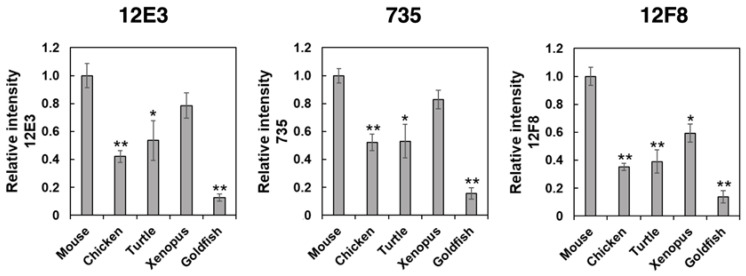
ELISA analysis of polySia-NCAM in brains from five different vertebrates. Brain homogenates (250 ng as protein) from five vertebrates (mouse, chicken, turtle, xenopus, goldfish, n = 3) were immobilized onto a 96-well plate and blocked with 2% BSA. The wells were then incubated with 12E3, 735, or 12F8 antibody before or after the Endo-N treatments. After color development, ELISA values of the Endo-N-treated wells were subtracted from those of the Endo-N-untreated wells. The left, middle, and right panels show the relative binding intensity of the 12E3, 735, and 12F8 antibodies, respectively. Triplicate analyses are shown with error bars. * indicates *p* < 0.05. ** indicates *p* < 0.01.

**Figure 6 ijms-21-08593-f006:**
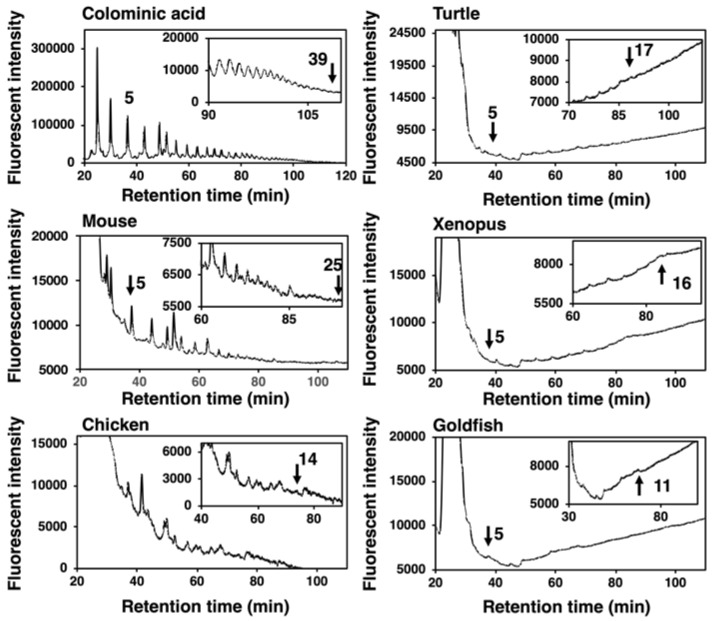
MH–FAEC analysis (Oligo-analysis) of brain homogenates from five vertebrate brains. Brains from five vertebrates (mouse, chicken, turtle, xenopus, goldfish, *n* = 1) were used. Brain homogenates (100 µg as protein) were subjected to mild acid hydrolysis followed by DMB derivatization. DMB-labeled oligo/polySia chains were applied to an anion exchange chromatography-HPLC analysis.

**Figure 7 ijms-21-08593-f007:**
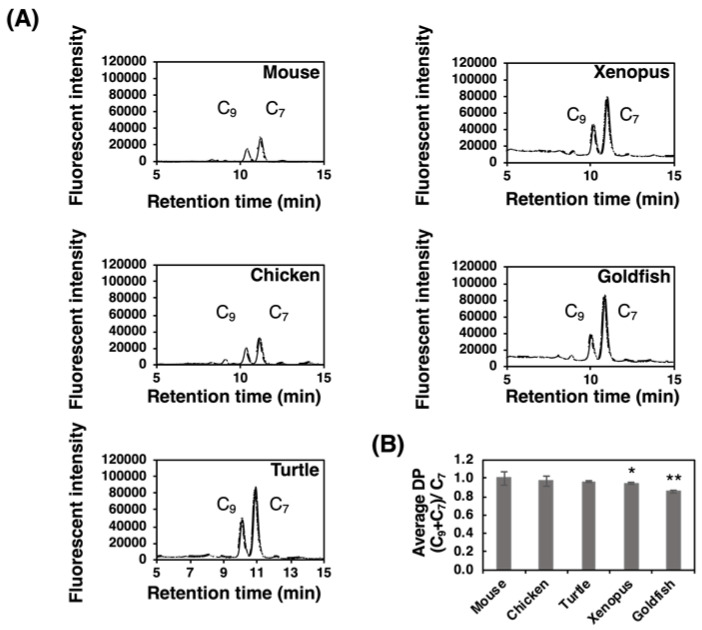
Fluorometric C_7_/C_9_ analysis of brain homogenates of five different vertebrate brains. Brains from five vertebrates (mouse, chicken, turtle, xenopus, and goldfish, *n* = 3) were used. Brain homogenates (20 µg as protein) were subjected to the fluorometric C_7_/C_9_ method to measure the average degree of polymerization (DP). (**A**) Typical chromatograms of C_7_/C_9_ analysis of each brain. (**B**) Relative (C_7_+C_9_)/C_7_ index in each brain. * indicates *p* < 0.05. ** indicates *p* < 0.01. Mouse value was set to 1.0.

**Figure 8 ijms-21-08593-f008:**
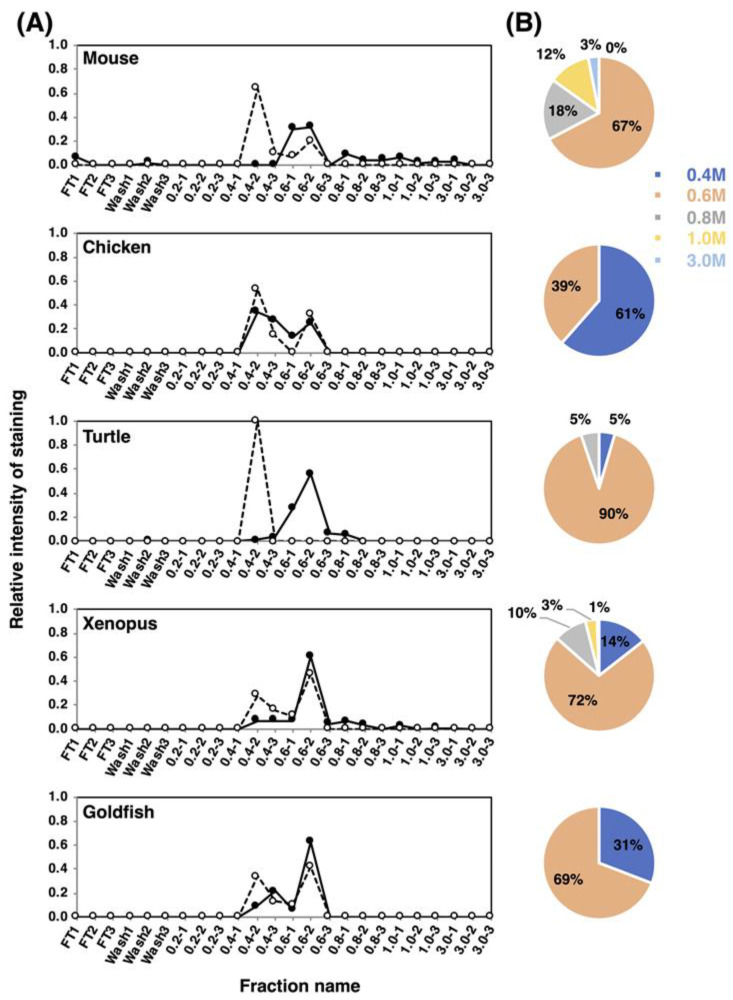
Net negative charge analysis of polySia-NCAM and endo-N-treated NCAM from brain homogenates of five vertebrate brains. Brains from five vertebrates (mouse, chicken, turtle, xenopus and goldfish, *n* = 1) were analyzed using DEAE-Sephadex A-25 anion-exchange chromatography. (**A**) Elution profiles of polySia-NCAM and endo-N-treated NCAM. Crude samples or endo-N-treated samples (500 μg as protein) were applied to the column, and eluted in a step-wise manner using 1.5 mL of 2 mM Tris-HCl (pH 8.0) buffer containing 0.1% Triton and NaCl (0.2, 0.4, 0.6, 0.8, 1.0, and 3.0 M). Then, 10 μL of the sample from each fraction was analyzed by SDS-PAGE/Western blotting. Crude samples were analyzed for the majority of the polySia on NCAM using the anti-polySia antibody 12E3 (solid line), while the Endo-N-treated samples, i.e., oligoSia-NCAM, were analyzed using the anti-NCAM antibody (broken line). Flow-through (FT) and wash fractions comprised only 0.1% Triton and 2 mM Tris-HCl (pH 8.0) with 0.15 M NaCl. The sum of all fraction intensities detected by Western blotting was set to 1.0. (**B**) Pie charts of the distribution of negative charges represented as NaCl concentration for each animal. Based on the chromatograms (left panels) of the crude sample, the intensity of 12E3 staining from each of the indicated NaCl concentrations was summed, and expressed as its proportions (%) to the summation of the intensities for all NaCl concentrations used. Blue, pink, grey, yellow, and sky blue represent 0.4 M, 0.6 M, 0.8 M, 1.0 M, and 3.0 M, respectively.

**Figure 9 ijms-21-08593-f009:**
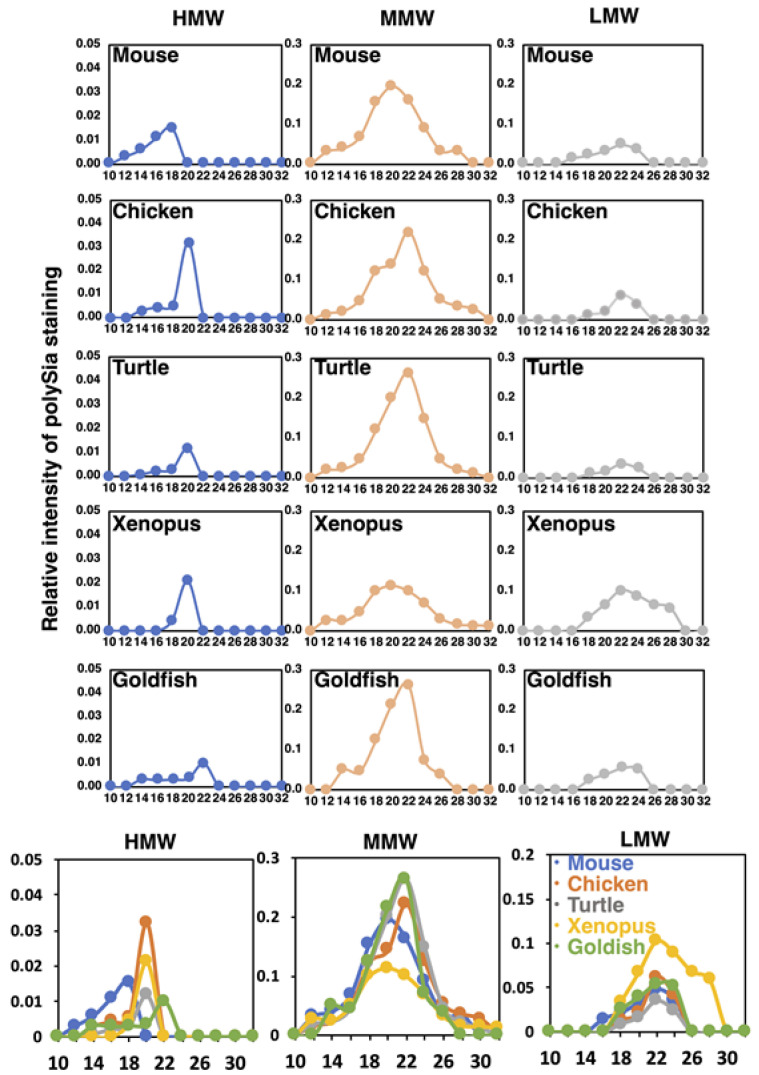
Sephacryl S-500 chromatograms of polySia-NCAM of brain homogenates from five vertebrate brains. Brains from five different vertebrates (mouse, chicken, turtle, xenopus, and goldfish, *n* = 1) were analyzed by the gel filtration. The elution profiles were monitored by native-PAGE/Western blotting of each fraction using 12E3 (Appendix A). The left, middle, and right panels for each animal brain show the elution profiles in the regions of HMW (above 720 kDa), MMW (480–720 kDa), and LMW (less than 480 kDa), respectively. The sum of the intensity of the immunostaining of the fractions was set to 1.0. At the bottom panels, merged images of the profiles for five different vertebrate brains are shown.

**Figure 10 ijms-21-08593-f010:**
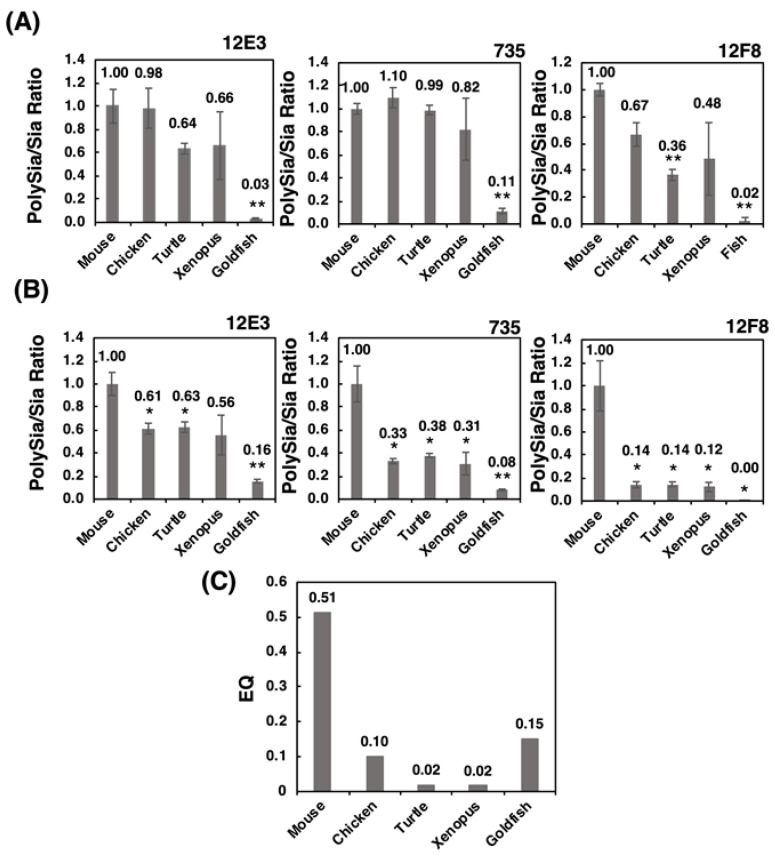
The polySia/Sia ratio and EQ value. The polySia/Sia ratio was calculated according to the intensity of anti-polySia-antibody per protein divided by the Sia amount per protein. (**A**) polySia/Sia ratio evaluated by SDS-PAGE/Western blotting. (**B**) PolySia/Sia ratio evaluated by native-PAGE/Western blotting. (**C**) The EQ value as a standard for evaluating the intelligence of vertebrates was calculated using an equation: EQ = K × [brain weight]/[body weight ]^3/4^ (K = 10, K was set to 10 for the convenience of calculation, although there were deviations among 8~16. * indicates *p* < 0.05. ** indicates *p* < 0.01.

**Figure 11 ijms-21-08593-f011:**
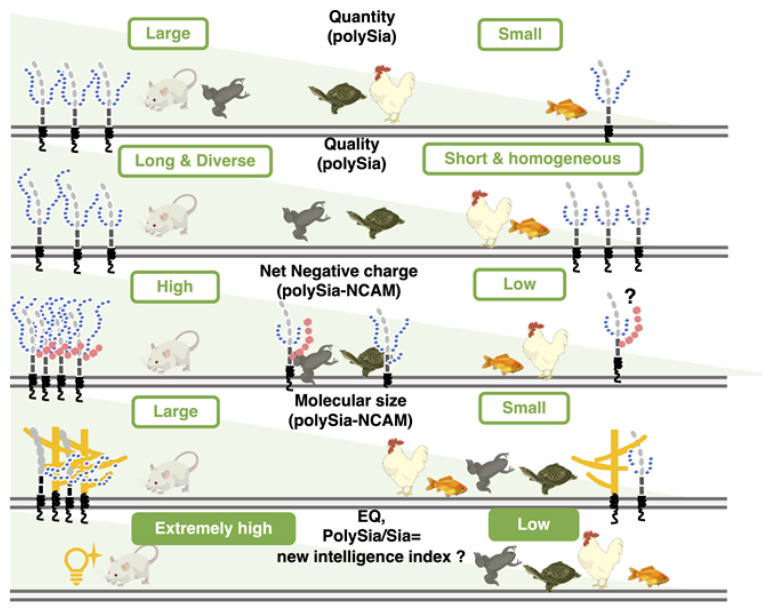
The quantity and quality of polySia and the features of polySia-NCAM in five different vertebrate brains. PolySia-NCAM is highly regulated in both quantity and quality (DP, net negative charge, and size) among the five different vertebrate brains.

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
