# Peer review of "Comparative Studies of Polysialic Acids Derived from Five Different Vertebrate Brains"

_ijms, 2020, doi:10.3390/ijms21228593_

Round 1

Reviewer 1 Report

The article covers an essential topic of Polysialic acid (polySia/PSA) and their role in brain evolutionary functions. The reviewed information provides detailed information about the composition of polySia/PSA derivates in 5 different vertebrates species. Illustrations are quite informative and accurately describes current knowledge. However, some minor comment should be corrected:

  • The manuscripts' discussion section presents graphs (fig. 10), encephalization quote, and "intelligence" index, but they should be presented as part of the results and explain what the results mean in the discussion
  • The authors hypothesize or propose a new index for intelligence, but the term for intelligence should be defined to provide more clarity. The term is vague. The intelligence could be a reflection of various task demands as recognition, memory, or neuronal efficiency. More discussion should be addressed
  • Although they are attractive results, some of them are preliminary results since they have an n=1, I suggest you complete their experiments or provide justifications about that. However, it should be better focus on discussing the results of their full experiments ( n=3)
  • Minor corrections: L398 and L399 there are typos, should be corrected

Author Response

Thank you for your comments and suggestions. We reply the comments.

Reviewer 2 Report

The authors adopted a newly developed combinational methods for the analyses and comparison of the structures of polySia-NCAM in brains of five different vertebrates. The results of this study suggest a positive correlation between molecular complexity and polySia and more advanced brain functions which likely reflect the complexity of brain function. They also hypothesized this parameter as a new promising index to evaluate the intelligence of different vertebrate brains. The manuscript is well written and results appear really interesting, however it requires major revisions for the reasons here exposed.  

In lines 179-180 the authors observed that “mouse brain only exhibited intense smears”, but WB images (Fig. 3) rather show similar smears more generally distributed among the species, however less evident in lower vertebrates because of the lower intensity in comparison to mouse. In addition, an experienced observer can easily see an artifact, an area of uneven protein blotting (often occurring during WB analyses). The shape of this artifact is present in all three WB showed, thus revealing also that several tests were performed using the same blot, I suppose after stripping. This could be responsible of additional problems or cross contaminations of results. I also suspect that reuse of the same blot was performed also in the SDS-PAGE experiments (the lanes shape of whole WBs in Fig. 2a and 2C are easily superimposable).

These observations can naturally affect also the results showed in fig.4. The differences in the slopes are mostly due to different intensities in WB, consequently the extension of the smears is underestimated. Hence, the statement: “polySia-NCAM in the brain was different not only in quantity (Fig. 2 and 3), but also in quality (Fig. 4)” should require a reevaluation because qualitative results are (due to the technique) underestimated by quantitative differences. The two analyses should be performed separately, or reevaluate the qualitative differences by normalization with an equal amount of polySia-NCAM per species.    

The ELISA results, as can be expected, are consistent with the amount of polySia in the native state than in the denatured state. These results are correctly indicative of different levels of polySia-NCAM in brains from the different vertebrates in the same 250 ng of protein extracts, an information already given by WBs.

The Authors further analyzed the brain sample by their own technique. Supernatant obtained by the ethanol precipitation of crude homogenates of vertebrate brains (100 μg as protein) were subjected to mild acid hydrolysis followed by DMB derivatization and applied to an anion exchange chromatography-HPLC analysis for subsequent separation of the labeled oligo/polySia in function of the degree of polymerization (DP). The Authors observed different DPs levels between the different vertebrates assuming that these data “were nearly consistent with those obtained by Western blotting” not considering that again, there is the same inaccuracy mixing quantitative with qualitative evaluations. In other terms, the longest polySia chains found chromatographically in mouse brain obtained after mild acid hydrolysis, reflect the higher amount of polySia in mice/total brain proteins. In other words, the more polySia, the more fragments from acid hydrolysis. The Authors should at least try to normalize the results of length analyses with a selected unit of polySia amount per species (12E3, 735, or 12F8). This can probably help to better evaluate the quality (DP) in the different species. Alternatively, the amount of each sample loaded on HPLC should be in function of the same polySia level, depending on the species.

Minor:

Line 126 African crowed flog (Xenopus laevis): “frog”

Author Response

(The authors gave the same response as above.)

Round 2

Reviewer 2 Report

no further comments